# Predictors of Oral Infection by Mucosal and Cutaneous Human Papillomaviruses in HIV-Infected and Uninfected Men Who Have Sex with Men of the OHMAR Study

**DOI:** 10.3390/jcm10132804

**Published:** 2021-06-25

**Authors:** Massimo Giuliani, Tarik Gheit, Francesca Rollo, Massimo Tommasino, Alessandra Latini, Maria Benevolo, Barbara Pichi, Raul Pellini, Sandrine McKay-Chopin, Antonio Cristaudo, Eugenia Giuliani, Aldo Morrone, Maria Gabriella Donà

**Affiliations:** 1STI/HIV Unit, San Gallicano Dermatological Institute IRCCS, 00144 Rome, Italy; massimo.giuliani@ifo.gov.it (M.G.); alessandra.latini@ifo.gov.it (A.L.); antonio.cristaudo@ifo.gov.it (A.C.); mariagabriella.dona@ifo.gov.it (M.G.D.); 2Infections and Cancer Biology Group, International Agency for Research on Cancer, 69372 Lyon, France; gheitt@iarc.fr (T.G.); tommasinom@iarc.fr (M.T.); chopins@iarc.fr (S.M.-C.); 3Pathology Department, Regina Elena National Cancer Institute IRCCS, 00144 Rome, Italy; maria.benevolo@ifo.gov.it; 4Otolaryngology Head&Neck Surgery Department, Regina Elena National Cancer Institute IRCCS, 00144 Rome, Italy; barbara.pichi@ifo.gov.it (B.P.); raul.pellini@ifo.gov.it (R.P.); 5Scientific Direction San Gallicano Dermatological Institute IRCCS, 00144 Rome, Italy; eugenia.giuliani@ifo.gov.it (E.G.); aldo.morrone@ifo.gov.it (A.M.)

**Keywords:** Human Papillomavirus, cutaneous, mucosal, oral cavity, HIV, men who have sex with men

## Abstract

Mucosal Human Papillomaviruses (HPVs) play a role in the development of a subset of head and neck cancers. Cutaneous HPVs are abundantly present in the oral cavity. The determinants of these infections have not been extensively investigated. We assessed the correlates of oral infection by alpha and beta and/or gamma HPVs in HIV-infected and uninfected men who have sex with men (MSM). Oral rinse-and-gargles were collected with a mouthwash. Alpha and beta/gamma HPVs were detected using the Linear Array HPV genotyping test and a multiplex PCR combined with Luminex technology, respectively. Multiple logistic regression was performed to identify independent predictors of oral HPV infection. Overall, 193 HIV-uninfected and 117 HIV-infected MSM were enrolled. Among HIV-infected MSM, the only determinant of alpha HPV infection was the number of lifetime oral sex partners (AOR: 8.26, 95% CI: 2.26–30.16). The strongest determinant of beta/gamma HPV infection was represented by practicing condomless receptive oral sex (AOR: 10.76, 95% CI: 1.56–74.17). Age was independently associated with alpha HPV infection in HIV-uninfected MSM. Beta/gamma HPV infection was not associated with sexual behavior in these subjects. In conclusion, predictors of oral infection differ between HIV-infected and uninfected MSM, as well as between alpha and beta/gamma HPVs.

## 1. Introduction

A small group of mucosal Human Papillomaviruses (HPVs) that belong to the alpha genus is involved in the etiology of oropharyngeal cancer. The HPV-attributable fraction globally amounts to 30% [1]. However, studies are increasingly reporting that cutaneous HPVs, mainly of the beta and gamma genera, are abundantly present in the oral cavity. The prevalence varies between 10 and 50% for beta HPVs [2,3,4,5,6], and is in the range of 3–30% for gamma types [3,4,6]. Despite the fact that ultimate data on the role of cutaneous HPVs in head and neck cancer (HNC) etiology are still lacking, oral infection by beta-1 HPV5, and gamma 11 and 12 species, has been demonstrated to be associated with incident HNC [7]. Independent studies have provided evidence that the epidemiologic determinants of mucosal and cutaneous HPV oral infection in the general population largely differ. Parameters of sexual behavior are among the predictors of alpha HPV infection [3,8,9,10], whereas conflicting findings have been reported for beta and gamma types [2,3,4]. Thus, additional studies are needed to thoroughly explore the determinants of infection with the different HPV genera. Compared with the general population, prevalence of oral infection by mucosal HPVs is higher in men who have sex with men (MSM), particularly in those living with HIV-1 infection [8,11,12,13,14,15]. In a previous study, we showed that beta and gamma HPVs are abundantly present in oral specimens from MSM, and that oral infection by beta HPVs is significantly more common than that caused by alpha types, irrespective of HIV status [6]. 

In the present study, we investigated the predictors of oral infection by mucosal (alpha) and cutaneous HPV (beta and gamma) in the same cohort of sexually active HIV-infected and uninfected MSM, analyzing socio-demographic, lifestyle, and sexual behavior characteristics.

## 2. Materials and Methods

### 2.1. Study Population and Data Collection

MSM (at least one same-sex sexual intercourse in the preceding 6 months) aged ≥18 years attending the STI/HIV Unit of the San Gallicano Dermatological Institute IRCCS (Rome, Italy) were recruited in the Oral/Oropharyngeal HPV in Men At Risk (OHMAR) study, which was conducted between November 2014 and February 2018. Enrollment criteria have been detailed previously [14]. Briefly, MSM without lesions suspicious for HNC (see below) were recruited if they reported no prophylactic HPV vaccination and no HNC history. Data on socio-demographic factors, lifestyle, lifetime, and recent (last 6 months) sexual behavior were collected through face-to-face interviews. Relevant data on the HIV-related parameters were retrieved from the medical records. A written informed consent was obtained from the study subjects. The study was approved by I.F.O. Section—Fondazione Bietti Ethics Committee (CE/417/14; approved on 13 May 2014). All procedures were performed in accordance with the Declaration of Helsinki.

### 2.2. Oral Clinical Evaluation

Expert otolaryngologists carried out an inspection of the oral cavity and oropharynx of each participant to ascertain the presence of lesions suspicious for cancer. They also rated the oral health of each study subject based on six parameters (presence of inflammatory status, caries, tartar, plaque, dental spots, and damaged or missing teeth). The scoring system defined by the otolaryngologists to categorize the oral health (as very good, good, fair, poor, and very poor) has been previously detailed [16].

### 2.3. Alpha, Beta and Gamma HPV Detection in Oral Samples

Oral rinse-and-gargles were collected as described previously [14]. Total nucleic acid extracts were obtained using the Amplilute Liquid Media Extraction Kit (Roche Molecular Diagnostics, Milan, Italy), following the manufacturer’s instructions. To detect alpha HPVs (37 genotypes), an aliquot was tested by the Linear Array HPV Genotyping Test (Roche Molecular Diagnostics). To detect beta (46 genotypes) and gamma HPVs (52 genotypes), type-specific assays developed at IARC (Lyon, France) were used [17].

### 2.4. Statistical Analysis

For the purposes of analysis, individuals were considered positive for mucosal HPVs when harboring at least one type of the alpha genus (any alpha), and for cutaneous HPVs when harboring at least one type of the beta and/or gamma genus (any beta/gamma).

Self-reported oral care was rated as good or poor, based on the responses given during the interview to six questions regarding gum bleeding, dental abscesses, toothache, tooth loss, frequency of dental cleaning, and mouthwash use, as detailed elsewhere [16].

Crude odds ratios (COR) and a 95% confidence interval (CI) were calculated by univariate analysis, which was performed separately, both for alpha and beta/gamma infection, and also for the two study groups (HIV-infected and uninfected MSM). Variables regarding oral sex always refer to receptive oral sex. Variables were coded as follows: (i) education: graduate vs. undergraduate; (ii) income: middle/upper (>12,000 euros/year) vs. low (≤12,000 euros/year); (iii) alcohol consumption: moderate/heavy (>1 dose per day) vs. no/light drinker (≤1 dose per day), where doses of alcoholic beverages were defined as described previously [14]; (iv) number of lifetime and recent partners for any and oral sex were categorized based on the respective median values; (v) condomless oral sex: yes (inconsistent condom use) vs. no (condom always used during intercourses); (vi) self-reported oral care: poor vs. good; (vii) clinician-rated oral health: poor (very poor/poor/fair) vs. good (good/very good). The multivariable model included age as a continuous variable, and all the covariates that showed a *p* < 0.10 at univariate analysis. When both indicators of any sex and oral sex behavior (collinear variables) were associated with the outcome of interest at univariate analysis, only those related to oral sex were included in the final model. Adjusted OR (AOR) and corresponding 95% CI were calculated. The statistical analyses were conducted using MedCalc Statistical Software version 19.3.1 (MedCalc Software Ltd., Ostend, Belgium; http://www.medcalc.org; 2020).

## 3. Results

### 3.1. Study Population

The study group was composed of 310 MSM. Of these, 117 (37.7%) were infected by HIV-1. Nadir and current CD4+ T-cell counts for HIV-infected MSM had a median of 300 (IQR: 202–403) and 636 cells/mm^3^ (IQR: 473–812), respectively. The large majority of these subjects were on combined antiretroviral therapy (cART) (110, 94.0%). With regard to socio-demographic features and lifestyle, the OHMAR study population has been previously described [15]. Briefly, HIV-infected and HIV-uninfected MSM differed significantly with respect to age (median age: 43 vs. 39 years) and number of recent oral sex partners (median of two vs. four). Moreover, HIV-uninfected individuals performed receptive oral sex (93.8% vs. 87.2%) and oral sex with occasional partners (85.6% vs. 73.5%) more frequently with respect to HIV-infected subjects (*p* < 0.05 for all comparisons). In addition, 40.2% of HIV-infected and 56.0% of HIV-uninfected MSM were non-smokers. None of the HIV-uninfected participants used pre-exposure prophylaxis (PrEP).

### 3.2. Correlates of Oral Infection by Alpha and Beta/Gamma HPVs

Of the 117 HIV-infected and 193 HIV-uninfected MSM, 28 (23.9%, 95% CI: 17.1–32.4) and 33 (17.1%, 95% CI: 12.4–23.0) harbored at least one alpha HPV genotype, respectively. Beta/gamma HPVs were detected in 75 (64.1%, 95% CI: 55.1–72.2) and 112 (58.0%, 95% CI: 51.0–64.8) individuals, respectively. Their prevalence significantly exceeded that of alpha HPVs in both study groups (*p* < 0.0001 both for HIV-infected and uninfected MSM).

Forest plots for the univariate analyses conducted to investigate the associations between oral infection by alpha and beta/gamma HPVs and selected variables are shown in Figure 1, separately according to HIV status.

Among HIV-infected MSM, oral infection by alpha HPVs showed significant associations with the number of lifetime (COR = 3.07, 95% CI: 1.18–7.94 for those with >95 vs. ≤95 partners, *p* = 0.018) and of recent partners for any sex (COR = 2.42, 95% CI: 1.01–5.78 for those with >5 vs. ≤5 partners, *p* = 0.044), as well as with the number of lifetime (COR = 8.43, 95% CI: 2.70–26.35 for those with >50 vs. ≤50 partners, *p* < 0.0001) and of recent partners for oral sex (COR = 2.90, 95% CI: 1.21–6.95 for those with >3 vs. ≤3 partners, *p* = 0.015) (Figure 1a). A borderline association was observed for smoking status (*p* = 0.08), and having occasional partners for any sex (*p* = 0.06).

For alpha and beta/gamma HPVs, odds of oral infection tended to increase with age in both study groups, but only for alpha HPVs in HIV-uninfected MSM did the association reach statistical significance (COR = 1.04, 95% CI: 1.01–1.08, *p* = 0.018) (Figure 1b). In the latter group of subjects, history of Sexually Transmitted Infections (STIs) showed a marginal association with alpha HPV infection (*p* = 0.065).

In HIV-infected MSM, borderline associations with beta/gamma HPV prevalence were observed for the number of recent partners for any sex (*p* = 0.06) and oral sex (0.096), having occasional oral sex partners (*p* = 0.06), practicing condomless oral sex (*p* = 0.054), and having a current CD4+ T-cell count <500 cells/mm^3^ (*p* = 0.056), as shown in Figure 1c. Among HIV-uninfected MSM, prevalence of beta/gamma HPVs increased significantly in those with a graduate education (COR = 2.33, 95% CI: 1.30–4.19, *p* = 0.004) and middle/upper income (COR = 1.80, 95% CI: 1.01–3.25, *p* = 0.047) (Figure 1d). A borderline association with condomless oral sex was also found (*p* = 0.087).

The results of multivariate analyses are shown in Table 1 and Table 2. For oral infection by alpha HPVs in HIV-infected MSM, an independent association was confirmed for the number of lifetime oral sex partners (Table 1). Odds of alpha HPV infection increased approximately eight times in those with >50 vs. ≤50 lifetime oral sex partners (AOR: 8.26, 95% CI: 2.26–30.16). Among the HIV-uninfected participants, only age was independently associated with prevalent alpha HPV infection with an increase in risk by 40% every 10 years of age (AOR: 1.04, 95% CI: 1.01–1.08), whereas the associations with the other variables entered in the model were not confirmed (Table 1).

In HIV-infected MSM, the odds of prevalent oral infection by beta/gamma HPVs were around 11 times as high for those who reported no condom use during receptive oral sex (AOR: 10.76, 95% CI: 1.56–74.17) (Table 2). Individuals with a current CD4+ T cell count <500 cells/mm^3^ had about 3-fold increased odds of having a prevalent beta/gamma HPV infection (AOR: 3.41, 95% CI: 1.14–10.18). Among HIV-uninfected MSM, those with graduate education (AOR: 2.19, 95% CI: 1.18–4.08) and middle/upper income (AOR: 2.09, 95% CI: 1.07–4.07) had higher odds of prevalent beta/gamma HPV infection (Table 2).

## 4. Discussion

Data about the prevalence of oral infection by alpha, beta, and gamma HPVs in the same groups of HIV-infected and uninfected MSM included in the present study have been previously presented and discussed [6]. Briefly, we found that oral HPV prevalence rates in our participants exceed those observed in non-MSM and other populations, possibly because of different demographic characteristics (e.g., age and ethnicity) and sexual behavior. Nonetheless, our alpha HPV prevalence is close to the pooled prevalence estimated for MSM in a relatively recent meta-analysis [13]. Alpha HPVs included in the current prophylactic vaccines have also been detected, and the possible implications of these findings for immunization have been previously discussed [18]. Regarding beta and gamma HPVs, the use of a type-specific assay with a wide capacity to detect multiple genotypes certainly may have contributed to the higher prevalence for these HPVs. Moreover, the prevalence rates observed for the three genera at oral level differed from those assessed at anal level, as already discussed [6,16]. 

Analyzing a wide spectrum of socio-demographic and behavioral characteristics, the present study focused on investigating the correlates of oral infection by mucosal and cutaneous HPVs separately for the two study groups. In order to compare the different genera, mucosal HPVs were analyzed as a whole, irrespective of their oncogenic potential or inclusion in the current prophylactic vaccines. It is also worth noting that epidemiological studies have extensively shown the broad distribution of beta and gamma HPVs at mucosal sites, so that this tropism-based classification may need to be revised. However, for the purposes of the present study and to simplify data presentation, we used the traditional assumption of alpha HPVs being mucosal, and beta and gamma HPVs being cutaneous types. 

The identified predictors appeared to be different for the two HPV groups, and to vary according to HIV status. Odds of alpha HPV infection significantly increased with the lifetime number of oral sex partners in HIV-infected MSM, whereas no significant associations with indicators of sexual behavior were found for the HIV-uninfected counterparts. We also observed a marginally significant increase in the odds of alpha HPV in HIV-infected MSM who were current smokers.

In HIV-uninfected subjects, odds of alpha HPV infection increased significantly with age. An increase of 40% every 10 years of age was observed. An association with older age has already been reported for HIV-uninfected individuals [19] and is consistent with a significant increase in the incidence of oral infection by alpha HPVs in older HIV-uninfected MSM of our cohort [16].

The strongest predictor of oral infection by alpha HPVs among HIV-infected MSM was represented by the number of lifetime oral sex partners. An association with this parameter has also been observed for heterosexual men [20], whereas others reported a relationship with recent, instead of lifetime, exposure through oral sex and confirmed the role of HIV as predictors of oral infection by alpha HPVs [21]. Among HIV-negative MSM, prevalent oral HPV infection showed a strong association with a higher number of recent oral sex partners [22]. These findings suggest a correlation between prevalent oral infection and oral sexual activity, but this can be revealed by different parameters of oral sexual behavior, related either to lifetime or recent exposure. This may also depend on the characteristics of the study population.

The strongest predictor of prevalent beta/gamma infection among HIV-infected MSM was the practice of condomless oral sex. This finding suggests that, at least in part, cutaneous HPVs may be acquired at oral level from the male genitals during unprotected receptive oral sex. Indeed, beta HPVs are widely represented in male genital skin, with a prevalence of about 80% [23,24]. Acquisition of cutaneous HPVs through oral sex on a man is also supported by the increased oral prevalence of beta HPVs in women with a higher number of lifetime oral sex partners [4]. Although it cannot be excluded that detection of cutaneous HPVs in oral samples is due to contamination from a partner’s genitals rather than a real infection, it must be noted that, for the beta/gamma HPV-positive individuals of this investigation, the median time since last oral sex was 10 days (data not shown). It is unlikely that contaminating cutaneous HPVs still reside in the oral cavity after this period, which is most likely sufficient for the salivary flow to eliminate merely deposited viral particles. Acquisition of cutaneous HPVs through autoinoculation or oro-anal contact cannot be excluded, but no information on rimming practices were available for our participants. Interestingly, an association with condomless oral sex was not observed for HIV-uninfected MSM, possibly because of the very low proportion of subjects reporting consistent condom use (2.8% of those who practiced receptive oral sex) and consequent low statistical power for this covariate.

The odds of beta/gamma HPV infection increased more than three times in HIV-infected MSM with a current CD4+ T-cell count <500 cells/mm^3^, suggesting that a poorer immunological status may favor the infection, either promoting its acquisition or delaying its clearance. An association of oral infection by mucosal HPVs with lower CD4 cell counts has been described by others [19,25], but, to the best of our knowledge, no data have been reported for cutaneous types.

No association between sexual behavior and beta/gamma oral infection was found in HIV-uninfected MSM. Similarly, oral infection by beta HPVs was not associated with parameters of sexual behavior for men of the HIM study, which were for the most part heterosexuals [24]. Overall, these findings suggest that sexual activity does not necessarily account for the presence of cutaneous HPVs in oral samples of both HIV-uninfected MSM and heterosexual men.

We did not observe a significant association of prevalent beta/gamma infection with age, either among HIV-infected or uninfected MSM, contrary to previous studies that found a significant association of beta HPV oral infection with older age [3,23]. These differences may derive from variation in the age structure of the study groups, but also from the HPV genus analyzed, given that no association with gamma HPVs with age was observed by Wong et al. [3]. However, it is also possible that our study was underpowered to evidence an association with age.

The limitations of the present study include: (i) only MSM were enrolled, thus our findings are not generalizable to men from the general population; (ii) the proportion of MSM in some categories was too low for meaningful analyses.

## 5. Conclusions

In conclusion, we found that predictors of oral infection differ between HIV-infected and uninfected subjects, as well as between alpha and beta/gamma HPVs. Lifetime number of oral sex partners was independently associated with alpha HPV infection in HIV-infected MSM, whereas only age showed an independent association with this infection in the HIV-uninfected counterparts. Condomless oral sex and lower current count of CD4+ T cells were independently associated with beta/gamma HPVs in HIV-infected MSM, whereas education and income were independent predictors of cutaneous HPV infection in the HIV-uninfected counterparts. Further studies are necessary to confirm the observed associations, in particular those with wide Cis and, thus, more uncertainty, to dissect the natural history and routes of transmission of cutaneous HPV oral infection, as well as its possible role in human disease.

## Figures and Tables

**Figure 1 jcm-10-02804-f001:**
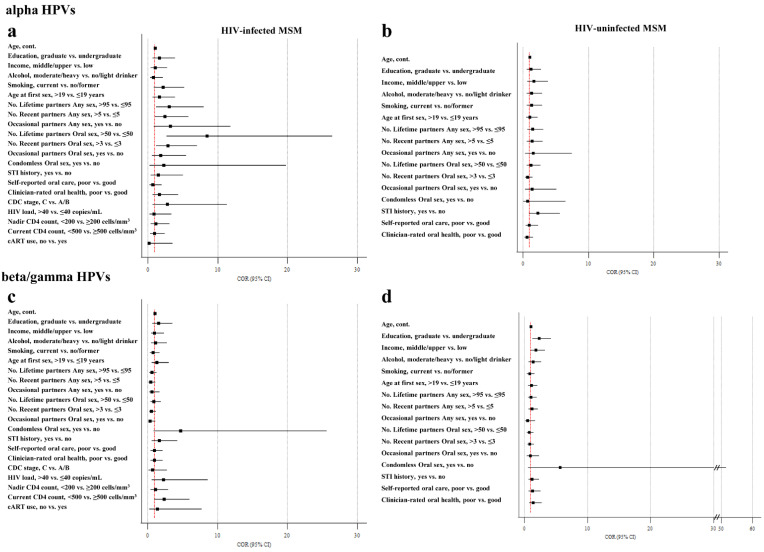
Forest plots of the univariate analyses for the associations with oral infection by (**a**,**b**) alpha HPVs and (**c**,**d**) beta/gamma HPVs in HIV-infected and uninfected MSM. The red line indicates the crude odds ratio (COR) for null hypothesis. CI, Confidence Interval.

**Table 1 jcm-10-02804-t001:** Multivariate analysis for prevalent oral infection by alpha HPVs in 310 MSM according to HIV status.

Variable	HIV-Infected MSM		HIV-Uninfected MSM	
	AOR (95% CI)	*p* Value	AOR (95% CI)	*p* Value
Age, years ^1^	1.00 (0.96–1.05)	0.90	1.04 (1.01–1.08)	0.04 *
Current smoker	2.38 (0.92–6.21)	0.07		
Occasional partners for any sex	1.95 (0.43–8.98)	0.39		
No. lifetime oral sex partners > 50 ^2^	8.26 (2.26–30.16)	0.001 *		
No. recent oral sex partners > 3 ^2^	1.00 (0.32–3.12)	0.99		
STI history ^3^			1.98 (0.80–4.92)	0.14

Covariates adjusted for included age and all variable with *p* < 0.10 at univariate analysis for each study group, separately. Adjusted Odds Ratios are thus shown only for the covariates included in the multivariate analysis. ^1^ Age was modeled as a continuous variable. ^2^ Oral sex always refers to receptive oral sex. ^3^ Syphilis, gonorrhea any site, genital herpes, anogenital warts. * Significant *p* values. AOR, Adjusted Odds Ratio. CI, Confidence Interval.

**Table 2 jcm-10-02804-t002:** Multivariable analysis for prevalent oral infection by beta/gamma HPVs in 310 MSM according to the HIV status.

Variable	HIV-Infected MSM		HIV-Uninfected MSM	
	AOR (95% CI)	*p* Value	AOR (95% CI)	*p* Value
Age, years ^1^	1.03 (0.98–1.08)	0.24	1.01 (0.98–1.04)	0.48
No. recent oral sex partners > 3 ^2^	0.75 (0.27–2.03)	0.57		
Occasional oral sex partners ^2^	0.38 (0.11–1.33)	0.13		
Condomless oral sex ^2^	10.76 (1.56–74.17)	0.016 *	4.75 (0.48–46.47)	0.18
Current CD4+ < 500 cells/mm^3^	3.41 (1.14–10.18)	0.028 *		
Graduate education			2.19 (1.18–4.08)	0.01 *
Middle/upper income			2.09 (1.07–4.07)	0.03 *

Covariates adjusted for included age and all variables with *p* < 0.10 at univariate analysis for each study group, separately. Adjusted Odds Ratios are thus shown only for the covariates included in the multivariate analysis. ^1^ Age was modeled as a continuous variable. ^2^ Oral sex always refers to receptive oral sex. * Significant *p* values. AOR, Adjusted Odds Ratio. CI, Confidence Interval.

## Data Availability

The data presented in this study are available on request from the corresponding author. The data are not publicly available due to the fact that they include personal data from vulnerable populations.

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
