# Peer review of "Predictors of Oral Infection by Mucosal and Cutaneous Human Papillomaviruses in HIV-Infected and Uninfected Men Who Have Sex with Men of the OHMAR Study"

_jcm, 2021, doi:10.3390/jcm10132804_

Round 1

Reviewer 1 Report

Authors have presented a manuscript entitled  “ Predictors of oral infection by mucosal and cutaneous Human 2 Papillomaviruses in HIV-infected and uninfected men who 3 have sex with men of the OHMAR study “ to be considered for publication in the journal of Clinical Medicine.

The article has an important topic regarding the prevalence and interaction of cutaneous HPV infections with mucosal alpha HPVs in oral cavity and their likely role in oral cancer development. However, for some reason the authors do not explore the HPV type-level co-occurrence or exclusion patters in this brilliant dataset of HIV-infected and HIV-negative MSM. Moreover, methods beyond pairwise comparison of HPV types should be used here and diversity patters should also be estimated. Finally, the authors could leverage their dataset analysis by comparing the presented broad HPVs type prevalence distribution to other anatomical sites broad HPV type prevalences among the HIV+/HIV- MSM.

Author Response

Please find below our point-by-point response to the reviewers’ comments.

 Reviewer: 1

Authors have presented a manuscript entitled “Predictors of oral infection by mucosal and cutaneous Human Papillomaviruses in HIV-infected and uninfected men who have sex with men of the OHMAR study “to be considered for publication in the Journal of Clinical Medicine. The article has an important topic regarding the prevalence and interaction of cutaneous HPV infections with mucosal alpha HPVs in oral cavity and their likely role in oral cancer development.

  1. However, for some reason the authors do not explore the HPV type-level co-occurrence or exclusion patters in this brilliant dataset of HIV-infected and HIV-negative MSM. Moreover, methods beyond pairwise comparison of HPV types should be used here and diversity patters should also be estimated.

Author response: We thank the reviewer for the comments and suggestions regarding further analyses of our dataset. However, the primary goal of this study was to explore the correlates of oral HPV infections by mucosal (alpha) and cutaneous HPVs (beta/gamma) in MSM by HIV status. Co-occurrence, exclusion or diversity patterns are certainly worth of investigation, but they would deserve an adequate description of the methods and results, as well as an adequate discussion. We feel that these results would not fit in with the present work and would require a different, dedicated paper.

  1. Finally, the authors could leverage their dataset analysis by comparing the presented broad HPVs type prevalence distribution to other anatomical sites broad HPV type prevalences among the HIV+/HIV- MSM.

Author response: Thank you for your suggestion about the opportunity to include in the manuscript a brief comparison of the prevalence of the three HPV genera at oral level with the prevalence at other anatomical sites. However, since prevalence data have been reported and widely discussed in a previous publication (included in the references), the required comparison has been only briefly mentioned at the beginning of the Discussion at line 211-213. 

Reviewer 2 Report

  1. Why are you using the alpha and beta groupings? These are heterogeneous as far as virulence. We really care about oncogenic and non-oncogenic and vaccine types. We also care about the specific HPV types like HPV16 and 18 which are the predominant causes of pharyngeal cancer in >85%.
  2. These correlations with sexual practices are of some interest but it is hard to know what this all means. Sara Oliver has published on these correlations. We know that HIV is a key factor.
  3. The prevalence here is vastly higher than the numbers published by others. Maybe you were testing for more types. About 10%-20% is the usual prevalence of oral HPV in men.
  4. What would be the impact of HPV immunization on the types you found?

Author Response

Please find below our point-by-point response to the reviewers’ comments.

 Reviewer: 2

  1. Why are you using the alpha and beta groupings? These are heterogeneous as far as virulence. We really care about oncogenic and non-oncogenic and vaccine types. We also care about the specific HPV types like HPV16 and 18 which are the predominant causes of pharyngeal cancer in >85%.

Author response: We agree with the reviewer that each genus includes HPVs that are very heterogeneous regarding carcinogenic potential, but also infection site and clinical manifestations they may cause. We also acknowledge that the classification in mucosal and cutaneous HPVs is somehow obsolete, especially considering the broad presence of so-called cutaneous types at mucosal sites. We have now highlighted these aspects at the beginning of the Discussion (lines 214-216 and lines 218-223).

Regarding oncogenic and vaccine types, the aim of the study was to evaluate the correlates of alpha HPV infection in comparison with beta/gamma HPVs, irrespective of the genotype oncogenic potential or inclusion in the HPV vaccine. In addition, the low number of HPV16 and HPV18 infections (n=11 and n=4, respectively, in the overall group of participants) and the scarce number of MSM positive for High-Risk HPVs in the two study groups (n=12 and n=21 among the HIV-infected and uninfected MSM, respectively) did not allow us to perform correlate-analyses for these genotypes, separately. To underline our study design, a phrase was added to the beginning of the Discussion at lines 216-218.   

  1. These correlations with sexual practices are of some interest but it is hard to know what this all means. Sara Oliver has published on these correlations. We know that HIV is a key factor.

Author response: We agree with the reviewer about the difficulty to fully understand the meaning of the associations observed between HPV oral infection and sexual practices. However, the scientific literature produced to date has widely confirmed these associations. The same study by Oliver et al., conducted on alpha HPVs, has highlighted the role of HIV infection and sexual behaviour in the acquisition of oral HPV. Our findings are consistent with those by Oliver et al, therefore we have now added a comment on this to the Discussion (lines 238-239) and included the article among the References (lines 394-396).

  1. The prevalence here is vastly higher than the numbers published by others. Maybe you were testing for more types. About 10%-20% is the usual prevalence of oral HPV in men.

Author response: Regarding alpha HPVs, our prevalence (23.9% and 17.1% in HIV-infected and uninfected MSM, respectively) is indeed higher than that observed in non-MSM but is close to the pooled prevalence estimated for MSM by a recent meta-analysis (King EM, Oomeer S, Gilson R, et al. PLoS One. 2016). Regarding beta and gamma HPVs, the use of a type-specific assay capable of detecting 46 beta genotypes and 52 gamma genotypes may certainly be one of the reasons why we observed a higher prevalence compared to other studies. Demographic characteristics (such as age, sex, ethnicity) as well as sexual behavior, may also contribute to explain the observed differences. All these aspects have been widely discussed in our previous publication that focused on the oral prevalence of alpha, beta and gamma HPVs in MSM (Gheit et al, Viruses. 2020 Aug 17;12(8):899). For this reason, we have avoided to discuss the results about the prevalence in the present paper. We have now added a brief discussion of the main findings regarding the prevalence data to the Discussion, recalling our previous paper for an adequate discussion of our findings (lines 201-206 and lines 209-211).

  1. What would be the impact of HPV immunization on the types you found?

Author response: Regarding the impact on immunization, this has been the object of our previous study (Donà MG, et al. Anal and oral human papillomavirus infection in men who have sex with men: implications for risk-targeted vaccination. Future Microbiol. 2020 Dec;15:1713-1722). For this reason, we have not discussed this topic in the present paper, which has a different focus. We have now added a sentence to the Discussion to recall our previous paper for an adequate discussion of the suggested topic (lines 207-209).

(Please see the attachment)

Reviewer 3 Report

This paper reports additional data on oral HPV infection in cohort of MSM, the main results of which have been published previously in several publications. Here the authors analyse predictors of infection, which are not yet well defined and are therefore of interest to the readership. The paper is well written and reflects and discusses current knowledge.

In this paper basic demographic characterists of the cohort are missing, the authors refer to previous publications for this information. They briefly summarize prevalence of HPV, but not the distribution of the variables which are being presented. It would be nice if the reader was briefly informed about age, proportion of study participants who smoke, have more than x sex partners, etc.. 
Is it known if any of the study participants have been vaccinated against HPV?

Table 1 and 2: is there a reason why selected/presented AORs are not shown for both HIV infected and uninfected MSM - rather than leaving gaps?

Discussion: The sentence "An increase by 40% every 10 years ..." in line 189 could be added to results in line 146.

Author Response

Please find below our point-by-point response to the reviewers’ comments.

  Reviewer 3

This paper reports additional data on oral HPV infection in cohort of MSM, the main results of which have been published previously in several publications. Here the authors analyse predictors of infection, which are not yet well defined and are therefore of interest to the readership. The paper is well written and reflects and discusses current knowledge.

  1. In this paper basic demographic characterists of the cohort are missing, the authors refer to previous publications for this information. They briefly summarize prevalence of HPV, but not the distribution of the variables which are being presented. It would be nice if the reader was briefly informed about age, proportion of study participants who smoke, have more than x sex partners, etc..

Authors’ response: a brief summary of the basic characteristics of the two study groups has been now provided at the beginning of the Results (lines 133-138).

  1. Is it known if any of the study participants have been vaccinated against HPV?

Authors’ response: As reported in M&M, MSM who had been vaccinated against HPV were not enrolled in the study (lines 79-81).

  1. Table 1 and 2: is there a reason why selected/presented AORs are not shown for both HIV infected and uninfected MSM - rather than leaving gaps?

Authors’ response: AORs are only shown for the covariates included in the multivariate analysis for the respective HPV genus and MSM group (e.g., No. lifetime oral sex partners only emerged as associated with alpha HPV infection for HIV-infected MSM at the respective univariate analysis and was thus included in the respective multivariate analysis; this covariate did not show any association with alpha HPV infection in HIV-uninfected MSM at the respective univariate analysis, thus it was not included in the respective multivariate analysis). We apologize if this was not clear. We have now added further details in this regard to the Table footnotes (lines 184-185 and lines 197-198).

  1. Discussion: The sentence "An increase by 40% every 10 years ..." in line 189 could be added to results in line 146.

Authors’ response: Thank you for the suggestion. The sentence has been added to the Results (line179).

(Please see the attachment)

Round 2

Reviewer 2 Report

No further comments

This manuscript is a resubmission of an earlier submission. The following is a list of the peer review reports and author responses from that submission.